# Antitumor Effect of Pyrogallol via miR-134 Mediated S Phase Arrest and Inhibition of PI3K/AKT/Skp2/cMyc Signaling in Hepatocellular Carcinoma

**DOI:** 10.3390/ijms20163985

**Published:** 2019-08-16

**Authors:** Hyojin Ahn, Eunji Im, Dae Young Lee, Hyo-Jung Lee, Ji Hoon Jung, Sung-Hoon Kim

**Affiliations:** 1College of Korean Medicine, Kyung Hee University, Seoul 02447, Korea; 2Department of Herbal Crop Research, National Institute of Horticultural and Herbal Science, Rural Development Administration (RDA), Eumseong 27709, Korea

**Keywords:** Pyrogallol, S-phase arrest, c-Myc, Skp2, miR-134

## Abstract

Though Pyrogallol, one of the natural polyphenols, was known to have anti-inflammatory and antitumor effects in breast and colon cancers, the underlying antitumor mechanisms of Pyrogallol, still remain unclear so far. Here, the antitumor mechanisms of Pyrogallol were elucidated in Hep3B and Huh7 hepatocellular carcinoma cells (HCCs). Pyrogallol showed significant cytotoxicity and reduced the number of colonies in Hep3B and Huh7 cells. Interestingly, Pyrogallol induced S-phase arrest and attenuated the protein expression of CyclinD1, Cyclin E, Cyclin A, c-Myc, S-phase kinase-associated protein 2 (Skp2), p-AKT, PI3K, increased the protein expression of p27, and also reduced the fluorescent expression of Cyclin E in Hep3B and Huh7 cells. Furthermore, Pyrogallol disturbed the interaction between Skp2, p27, and c-Myc in Huh7 cells. Notably, Pyrogallol upregulated miRNA levels of miR-134, and conversely, miR-134 inhibition rescued the decreased expression levels of c-Myc, Cyclin E, and Cyclin D1 and increased the expression of p27 by Pyrogallol in Huh7 cells. Taken together, our findings provide insight that Pyrogallol exerts antitumor effects in HCCs via miR-134 activation-mediated S-phase arrest and inhibition of PI3K/AKT/Skp2/cMyc signaling as a potent anticancer candidate.

## 1. Introduction

Liver cancer is recognized as the sixth most common cancer and the fourth leading cause of cancer mortality all over the world [1]. Among them, hepatocellular carcinoma (HCC) accounts for almost 90% of all primary hepatic malignancies [2]. Although chemotherapy, mainly with sorafenib, radiotherapy, surgery, and immunotherapy have been used for the treatment of HCC for years, the therapies are unsatisfactory due to its high recurrence rates and frequent accompanying cirrhosis [3,4]. Therefore, combination treatments to overcome resistance to anticancer agents were recommended for effective cancer therapy [5,6]. Consistently, Fan et al. reported the synergistic effect of 3-bromopyruvate as an effective glycolytic inhibitor in combination with other antitumor drugs (e.g., cisplatin, doxorubicin, 5-fluorouracil) in HL-60, U937, and HCT116 cells [7].

It is well documented that cell cycle arrest, including G0/G1, S, and G2/M phases are another option regardless of apoptosis induction in cancers [8,9]. Among them, many natural products such as lupeol [10], Biochanin A [11], and sulforaphane [12] are attractive for targeting S-phase arrest.

MicroRNAs (miRNAs), endogenous small non-coding RNAs of 19–25 nucleotides in length, are well known to act as a tumor suppressors or oncogenes in several cancers by regulating gene expression at a post-transcriptional level [13,14,15]. Among them, miRNA 134 was reported to act as a tumor suppressor in breast cancer [16], lung cancer [17], endometrial cancer [18], gastric cancer [19], osteosarcoma [20], and glioma cancer [21]. 

Pyrogallol, a major compound contained in *Emblica officinalis* and *Mangifera indica*, was known to have anti-inflammatory [22] and antitumor effects in colon cancer [23], breast cancer [24], leukemia [25] and lung cancer [26,27]. Nonetheless, the underlying antitumor mechanisms of Pyrogallol have not been elucidated in hepatocellular carcinoma yet. Thus, in the present study, the antitumor mechanisms of Pyrogallol were investigated in Hep3B and Huh7 HCCs in association with S-phase arrest and miRNA134 upregulation.

## 2. Results

### 2.1. Pyrogallol Reduced the Viability and Proliferation of Hep3B and Huh7 Cells

The effect of Pyrogallol (Figure 1A) was evaluated on the viability and proliferation of two hepatocellular carcinoma cells (Hep3B and Huh7) by the MTT assay. After cells were treated with various concentrations of Pyrogallol (0, 5, 10, 20, 40, 80 μM) for 24 h, MTT and colony formation assays were conducted. The MTT assay revealed that Pyrogallol decreased the viability of Hep3B and Huh7 in a dose-dependent manner (Figure 1B). Furthermore, Pyrogallol reduced the number of colonies in Hep3B and Huh7 cells by the colony formation assay (Figure 1C).

### 2.2. Pyrogallol Induced S-phase Arrest in Hep3B and Huh7 Cells

To identify the effect of Pyrogallol on the cell cycle phases in Hep3B and Huh7 cells, the cells treated with various concentrations of Pyrogallol were subjected to cell cycle analysis by using flow cytometry after staining with PI. Herein, S-phase arrest was induced by Pyrogallol in Hep3B and Huh7 cells (Figure 2A,B).

### 2.3. Pyrogallol Regulated the Expression of Cell Cycle-Related Proteins in Hep3B and Huh7 Cells

Cell cycle analysis revealed that Pyrogallol induced S-phase arrest in Hep3B and Huh7 cells. Consistently, Pyrogallol attenuated the protein expression of Cyclin D1, Cyclin E, and Cyclin A (Figure 3A) and also reduced the fluorescent expression of Cyclin E in Hep3B and Huh7 cells (Figure 3B).

### 2.4. Pyrogallol Decreased the Expression of c-Myc, Skp2, p-AKT, PI3K and Increased the expression of p27 in Hep3B and Huh7 Cells

Skp2 (S-phase kinase-associated protein 2) is involved in tumorigenesis, regulation of cell cycle and cell survival by targeting c-Myc and p27 [28,29]. Here the effect of Pyrogallol was assessed on Skp2, c-Myc, PI3K, and p-AKT in Hep3B and Huh7 cells by Western blotting. We found that the expression of Skp2, c-Myc, PI3K, and p-AKT was attenuated, but p27 was upregulated in Pyrogallol-treated Hep3B and Huh7 cells (Figure 4).

### 2.5. Pyrogallol Reduced the Expression of Skp2 and Disturbed the Interaction Between Skp2, p27, and c-Myc in Huh7 Cells

To check the effect of Pyrogallol on Skp2, immunofluorescence was conducted in Hep3B and Huh7 cells. Here Pyrogallol decreased the expression of Skp2 in Hep3B and Huh7 cells (Figure 5A). To confirm the direct binding between Skp2, p27, and c-Myc, since the binding score between Skp2 and p27, p27 and c-Myc, Skp2 and c-Myc was 9, 9, and 8, respectively by the STRING database (Figure 5B), the immunuoprecipitation assay was conducted in Hep3B and Huh7 cells. Immunoprecipitation confirmed that the interaction between Skp2, p27, and c-Myc was disturbed by Pyrogallol in Huh7 cells (Figure 5C).

### 2.6. The Pivotal Role of miR-134 in Pyrogallol-Induced S-Phase Arrest and Antiproliferation in Huh7 Cells

It is well documented that miR-134 was highly expressed in lung tumor, pancreatic cancer, colon cancer, and prostate cancer while it was minimally expressed in glioblastomas, breast cancer, renal cell carcinoma, colorectal cancer, hepatocellular carcinoma, and osteosarcoma cell lines [30]. To demonstrate the important role of miR-134 in Pyrogallol-treated Huh7 cells, RT-qPCR analysis was performed in Huh7 cells. The mRNA level of miR-134 was significantly induced by Pyrogallol in Huh7 cells (Figure 6A). However, miR-134 inhibition rescued the decreased expression levels of c-Myc, Cyclin E, and Cyclin D1 and increased the expression of p27 by Pyrogallol in Huh7 cells (Figure 6B).

## 3. Discussion

Due to bad prognosis and unsatisfactory efficacy in HCC treatment, recently natural compounds such as aloperine [31], 7-deoxynarciclasine [32], quercetin [33], avicularin [34], and decursin [35] have been attractive in the treatment of HCC. On the same line, here the underlying antitumor mechanism of Pyrogallol contained in *Emblica officinalis* and *Mangifera*
*indica* was explored in Hep3B and Huh7 HCCs in association with S-phase arrest and miR-134 activation.

Pyrogallol significantly increased cytotoxicity and reduced the number of colonies in Hep3B and Huh7 cells, implying cytotoxic and antiproliferative effects of Pyrogallol. 

It is well documented that cell proliferation depends on four distinct phases of the cell cycle including G0/G1, S, G2, and M, which is usually regulated by several cyclin-dependent kinases (CDKs) and so dysregulation of the cell cycle is regarded a promising target for cancer therapy [36]. Here, cell cycle analysis revealed that Pyrogallol induced S-phase arrest, while not affecting sub G1 population, in Hep3B and Huh7 cells. To confirm S-phase arrest induced by Pyrogallol, Western blotting was conducted in Hep3B and Huh7 cells targeting S-phase-related proteins. Herein Pyrogallol induced downregulation of CyclinD1, Cyclin A, and Cyclin E, and upregulation of p27 and also reduced the fluorescent expression of Cyclin E in Hep3B and Huh7 cells, indicating S-phase arrest by Pyrogallol. 

S-phase kinase-associated protein 2 (Skp2) regulates phosphorylated cell cycle regulator proteins and their ubiquitination as a cell cycle regulator and oncogene [29,37], c-Myc also works as an oncogene and cell cycle regulator [38], while PI3K and AKT are considered typical survival proteins in several cancers [39]. As expected, Pyrogallol attenuated the expression of Skp2, c-Myc, and PI3K/pAKT in Hep3B and Huh7 cells, demonstrating antiproliferative effects of Pyrogallol via inhibition of Skp2/ c-Myc and PI3K/ pAKT signaling. Similarly, Nemec et al. reported that Pyrogallol increased phosphorylation of AMPK and decreased phosphorylation of the AKT/mTOR pathway in MCF10DCIS.com breast cancer cells.

Emerging evidence reveals that regulation of protein protein interaction(PPI) networks is considered as a hot strategy for cancer therapy [40,41]. Interestingly, we found that Pyrogallol disturbed the interaction between Skp2, p27, and c-Myc in Huh7 cells, implying that the antitumor effect of Pyrogallol is mediated by the inhibition of PPI between Skp2, p27, and c-Myc.

Of note, Pyrogallol significantly increased the expression level of miR-134 that was known to be a tumor suppressor [14,19], since miR-134 was minimally expressed in HepG2, Hep3B, and SMMC7721 HCCs compared to L02 l normal liver cell lines [42]. Conversely, inhibition of miR134 rescued c-Myc and cyclin E rather than p27 or cyclin D1 in Pyrogallol-treated Huh 7 cells, implying that miR134 activation mediates the antitumor effect of Pyrogallol at least in part, which requires further study on the detailed mechanisms of Pyrogallol in HCCs in the future.

In summary, Pyrogallol showed significant cytotoxic and antiproliferative effects, induced S-phase arrest, attenuated the protein expression of CyclinD1, Cyclin E, Cyclin A, c-Myc, Skp2, p-AKT, PI3K, increased p27, reduced the fluorescent expression of Cyclin E, and disturbed the interaction between Skp2, p27, and c-Myc in Huh7 cells. Notably, Pyrogallol upregulated miRNA levels of miR-134 and conversely miR-134 inhibition rescued the decreased expression levels of c-Myc, Cyclin E, and Cyclin D1 and increased expression of p27 by Pyrogallol in Huh7 cells. Overall, our findings suggest the scientific evidence that Pyrogallol exerts antitumor effecst in HCCs via S-phase arrest and miR-134 activation as a potent anticancer agent (Figure 7).

## 4. Materials and Methods

### 4.1. Pyrogallol and Reagents

Pyrogallol (CAS No. 87-66-1), 3-(4,5-dimethylthiazol-2-yl)-2,5-diphenyltetrazolium bromide (MTT) and beta-actin were purchased from Sigma (Sigma, St. Louis, MO, USA). Also, specific antibodies for Skp2, CyclinE, CyclinA, p27 (Santa cruz Biotechnology, Dallas, USA), c-Myc (abcam, Cambridge, UK), cyclinD1, p-AKT, and PI3K (Cell signaling Technology, Danvers, MA, USA) were purchased for Western blot analysis.

### 4.2. Cell Culture

The human hepatocellular carcinoma cell line such as Hep3B (ATCC^®^ HB-8064^™^, Manassas, VA, USA was purchased from American Type Culture Collection (ATCC, Manassas, VA, USA) and the Huh7 cell line was bought from the Korean Cell Line Bank (KCLB, Seoul, Korea). Hep3B cells were cultured in DMEM with 10% FBS and 1% antibiotics and Huh7 cells were maintained in RPMI1640 (Welgene, Gyeongsan, Korea) in a humidified atmosphere of 5% CO_2_ at 37 °C.

### 4.3. Cell Viability Assay

The effect of Pyrogallol on the viability of the Hep3B and Huh7 cells was measured by 3-(4,5- dimethylthiazol-2-yl)-2,5-diphenyltetrazolium bromide (MTT) assay. Hep3B and Huh7 cells were seeded in a 96-well plate for 24 h and were exposed to various concentrations (0, 5, 10, 20, 40, 80 μM) of Pyrogallol for 24 h, then added to the MTT solution (1 mg/mL) and incubated at 37 °C for 2 h. After formazan was dissolved in DMSO, optical density was measured using a microplate reader (Molecular Devices Co., Silicon Valley, CA, USA) at 570 nm. Cell viability was calculated as a percentage of viable cells in the Pyrogallol-treated group versus the untreated control.

### 4.4. Colony Formation

Hep3B and Huh7 cells (3 × 10^3^/well) were seeded onto six-well plates and incubated at 37 °C. After 24 h, the cells were exposed to the concentrations (0, 20, 40, 80 μM) of Pyrogallol for 24 h and culture medium was replaced by a fresh one. After 10–12 days, cells were washed PBS, fixed, and stained with Diff quick solution (Sysmex, Kobe, Japan). The plate was dried at room temperature and colonies were manually counted under light microscopy.

### 4.5. Cell Cycle Analysis

Hep3B and Huh7 cells were treated with Pyrogallol (0, 20, 40, 80 μM) for 24 h, washed with PBS and fixed in 70% ethanol overnight at −20 °C. The cells were washed with cold PBS and treated with RNase A (1 mg/mL) for 40 min at 37 °C and stained PI (50 μg/mL) at room temperature and were analyzed using the FACS caliber (Becton Dickinson, Franklin Lakes, NJ, USA).

### 4.6. Real-time Quantitative PCR (RT-qPCR)

Total RNAs were extracted from Hep3B and Huh7 cells by using the QIAzol Lysis Reagent (QIAGEN, Hilden, Germany). The cDNA was synthesized with oligodT (bioneer) and M-MLV reverse transcriptase (Enzynomics, Gyeonggi-do, Korea). qRT-PCR was conducted according to Light cycler TM manufacturer’s protocol. The sequence of miR-134 was as follows: 5′-UGUGACUGGUUGACCAGAGGGG-3′

### 4.7. Western Blotting

Hep3B and Huh7 cells (2 × 10^5^/well) treated with Pyrogallol (20, 40, 80 μM) for 24 h were lysed with RIPA buffer containing protease inhibitor and phosphatase inhibitor for 30 min on ice. Lysate were centrifuged at 13,000× *g* for 25 min. Proteins were separated on 8%–12% SDS-PAGE gel and transferred on to Nitrocellulose transfer membrane (GE healthcare, Chicago, IL, USA) at 300 mA. After blocking with 5% non-fat skim milk in TBST for 1 h and probed with antibodies for Skp2, Cyclin E, Cyclin A, p27 (Santa Cruz Biotechnology, Dallas, TX, USA), beta-actin (sigma, St. Louis, MI, USA), c-Myc (Abcam, Cambridge, UK), cyclin D1, p-AKT, and PI3K (Cell signaling Technology, Danvers, Essex County, MA, USA).

### 4.8. Immunofluorescence

Hep3B and Huh7 cells were seeded onto four-well slide and exposed to 80 μM Pyrogallol for 24 h at 37 °C. The cells were fixed 4% paraformaldehyde and permeabilized with 0.1% Triton-X 100 in PBS for 10 min and incubated with 5% bovine serum albumin (BSA) for 1 h. After washing with PBS, the plate was probed with Cyclin E and Skp2 antibody overnight at 4 °C, and then with FITC secondary antibody for 1 h. The cells were stained with DAPI and mounted in medium (Vector Laboratories, Burlingame, CA, USA), and visualized under the FLUOVIEW FV10i confocal microscope (Olympuse, Japan).

### 4.9. Immunoprecipitation

The cells were lyzed in NP-40 lysis buffer, and then 300 μg of proteins were incubated overnight with antibody c-Myc at 4 °C and followed by protein A/G Plus-beads (Santa Cruz Biotechnology, Dallas, TX, USA) overnight at 4 °C and washed with NP-40 buffer. Immunoprecipitants were subjected to immunoblotting with the indicated antibodies.

### 4.10. MicroRNA Transfection

The miR-134 inhibitor and miR control (40 nM) (Bioneer, Daejeon, Korea) were transfected into Huh7 cells using the X-treme GENE HP DNA Transfection reagent (Roche, Basel, Switzerland) according to the manufacture’s protocol.

### 4.11. Statistical Analysis

All data were expressed as means ± standard deviation (SD) and the Student *t*-test was applied by using sigma plot for the comparison of groups. Values of *p* < 0.05 was considered a significant difference between groups.

## 5. Conclusions

Our findings suggest that Pyrogallol showed significant cytotoxicity and reduced the number of colonies in Hep3B and Huh7 cells. Interestingly, Pyrogallol induced S-phase arrest and attenuated the protein expression of Cyclin D1, Cyclin E, Cyclin A, c-Myc, S-phase kinase-associated protein 2 (Skp2), p-AKT, PI3K, increased the protein expression of p27, and also reduced the fluorescent expression of Cyclin E in Hep3B and Huh7 cells. Furthermore, Pyrogallol disturbed the interaction between Skp2, p27, and c-Myc in Huh7 cells. Notably, Pyrogallol upregulated miRNA level of miR-134 and conversely miR-134 inhibition rescued the decreased expression levels of c-Myc, Cyclin E, and Cyclin D1 and increased expression of p27 by Pyrogallol in Huh7 cells. Taken together, our findings provide insight that Pyrogallol exerts antiproliferative effects in HCCs via miR-134 activation mediated S-phase arrest and inhibition of PI3K/AKT/Skp2/cMyc signaling as a potent anticancer candidate.

## Figures and Tables

**Figure 1 ijms-20-03985-f001:**
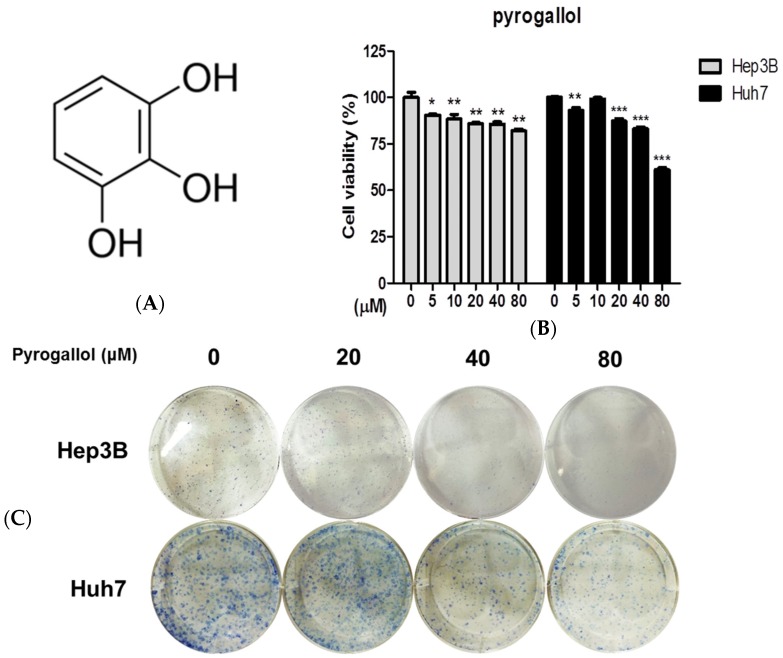
Pyrogallol decreased the viability of Hep3B and Huh7 cells. (**A**) Chemical structure of Pyrogallol. (**B**) Cells were seeded in a 96-well plate and exposed to various concentrations of Pyrogallol (0, 5, 10, 20, 40, 80 μM) for 24 h and cell viability was determined by MTT assay. (**C**) Photos for colony formation of Pyrogallol (0, 5, 10, 20, 40, 80 μM) in Hep3B and Huh7 cells. The colonies were visualized by staining with crystal violet. Data are expressed as means ± SD. * *p* < 0.05, ** *p* < 0.01, *** *p* < 0.001.

**Figure 2 ijms-20-03985-f002:**
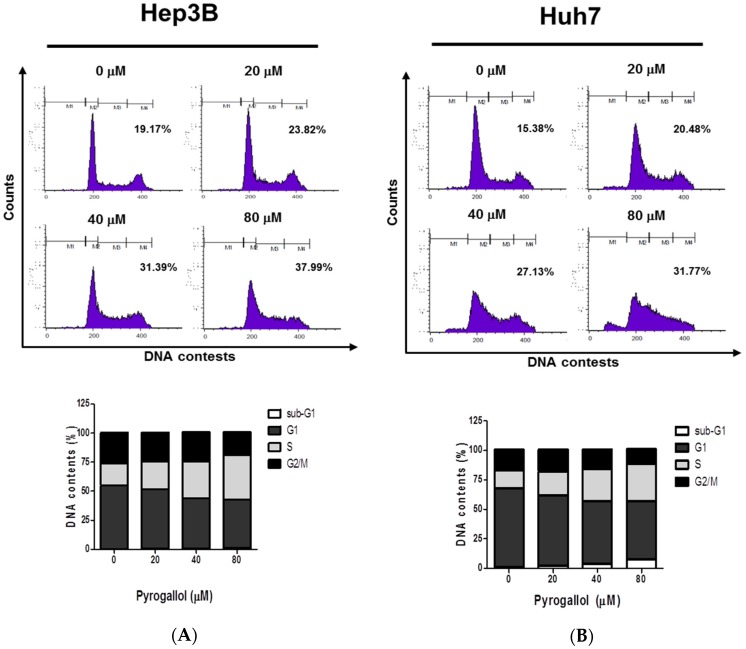
Pyrogallol induced S-phase arrest in Hep3B and Huh7 cells. (**A**,**B**) Hep3B and Huh7 cells were treated with Pyrogallol (0, 20, 40, 80 μM) for 24 h. The cells were washed with PBS, fixed in 70% ethanol, stained with PI and then subjected to flow cytometric analysis. Graphs show each phase of the cell cycle population (%).

**Figure 3 ijms-20-03985-f003:**
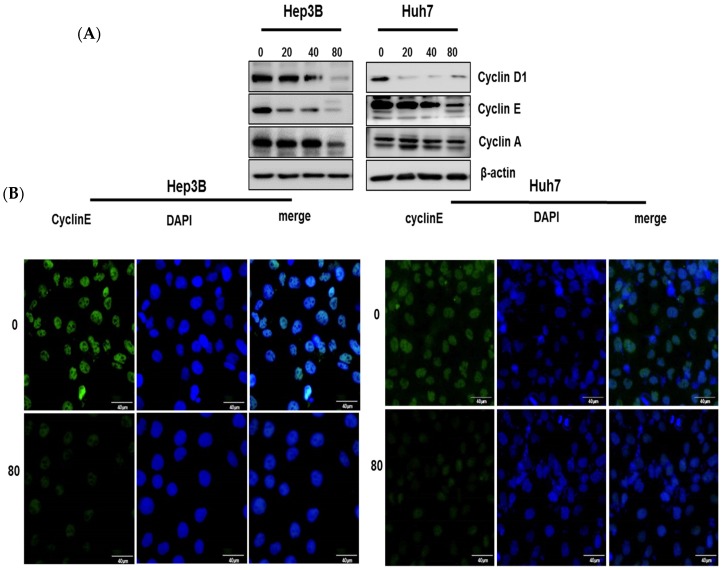
Pyrogallol regulated the expression of cell cycle-related proteins in Hepatocellular carcinoma. (**A**) Hep3B and Huh7 cells were treated with Pyrogallol (0, 20, 40, 80 μM) for 24 h. Significantly decreased expression of cell cycle-related protein cyclin D1, Cyclin E, and Cyclin A by Western blotting. (**B**) Cells were treated with 80 μM of Pyrogallol for 24 h and fixed, permeabilized, probed with Cyclin E antibody and secondary (FITC) antibody and were stained with DAPI and mounted in media and visualized under the FLUOVIEW FV10i confocal microscope (Olympuse, Japan). Scale bars = 40 μm.

**Figure 4 ijms-20-03985-f004:**
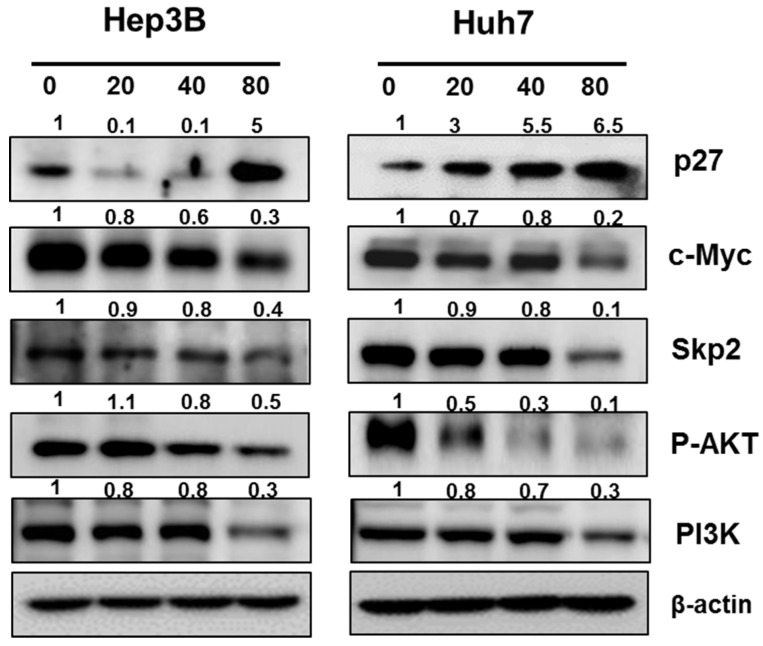
Pyrogallol modulated the expression of p27, c-Myc, Skp2, p-AKT and PI3K in Hep3B and Huh7 cells.

**Figure 5 ijms-20-03985-f005:**
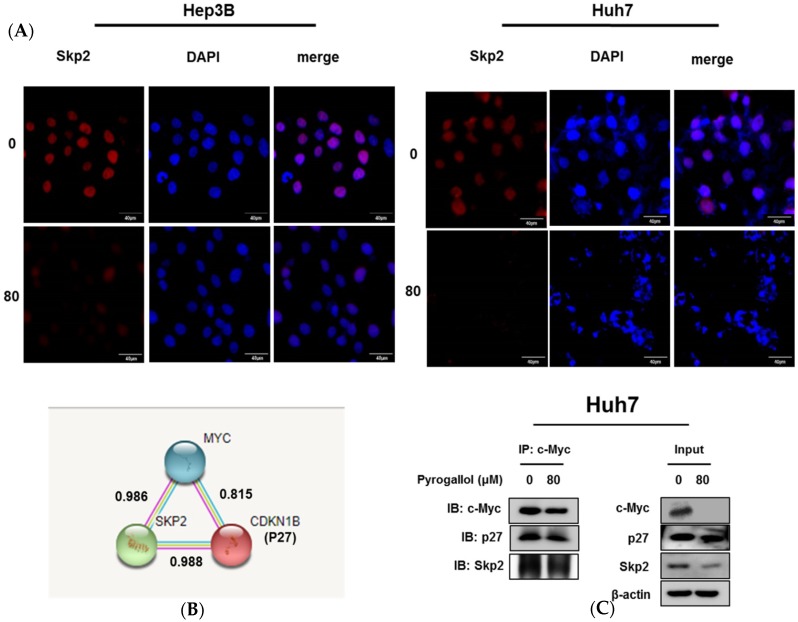
Pyrogallol reduced the expression of Skp2 and disturbed the interaction between Skp2 and c-Myc in Huh7 cells. (**A**) Effect of Pyrogallol on Skp2 expression in Hep3B and Huh7 cells. The cells were treated with 80 μM Pyrogallol for 24 h, fixed, permeabilized, and then probed with Skp2 antibody and with secondary (FITC) antibody and were stained with DAPI and mounted in media, and visualized under the FLUOVIEW FV10i confocal microscope (Olympuse, Japan). Scale bars = 40 μm. (**B**) Interaction Network scores between c-Myc and Skp2 by STRING. (**C**) Effect of Pyrogallol on the interaction between Skp2, p27, and c-Myc in Huh7 cells. Immunoprecipitation was performed in Huh7 cells by using c-Myc, p27, and Skp2 antibodies and then was subjected to Western blotting.

**Figure 6 ijms-20-03985-f006:**
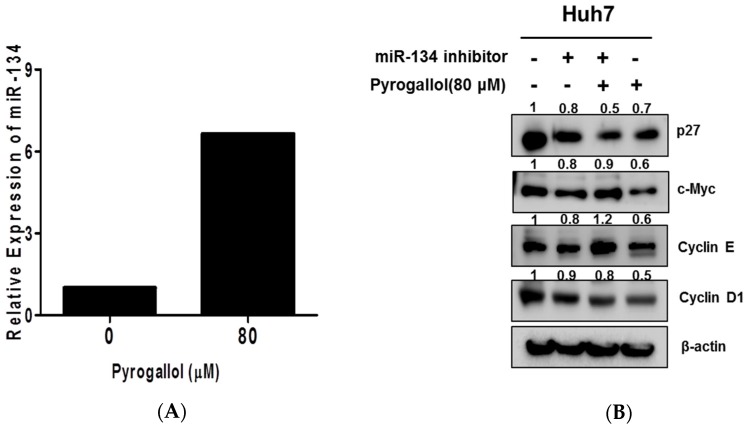
The pivotal role of miR-134 in Pyrogallol-regulated cell cycle-related proteins in Huh7 cells. (**A**) RT-qPCR analyzed the expression of miR-134 levels in Huh7 cells. (**B**) Huh7 cells were transfected with miR-134 for 48 h and exposed to Pyrogallol for 24 h. The expression levels of p27, c-Myc, Cyclin E, and Cyclin D1 were evaluated by Western blotting.

**Figure 7 ijms-20-03985-f007:**
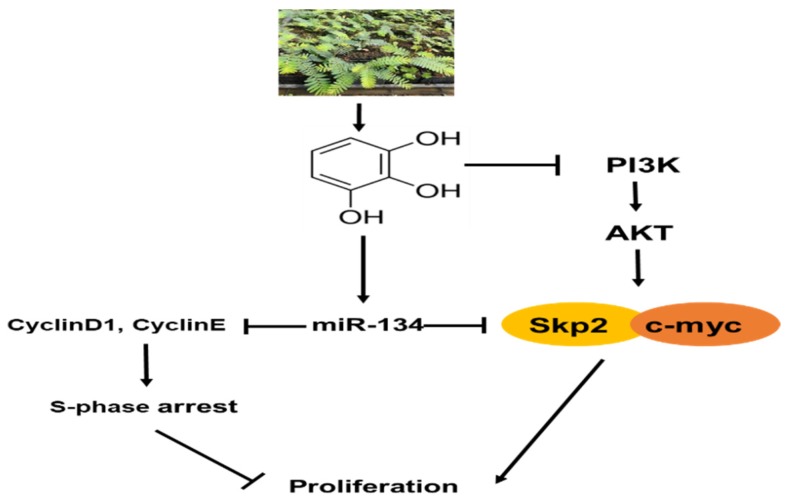
Molecular mechanisms of Pyrogallol via miR-134 activation mediated S-phase arrest and inhibition of the P13K/AKT/SKP2 and cMyc signaling axis.

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
