# Peer review of "Antitumor Effect of Pyrogallol via miR-134 Mediated S Phase Arrest and Inhibition of PI3K/AKT/Skp2/cMyc Signaling in Hepatocellular Carcinoma"

_ijms, 2019, doi:10.3390/ijms20163985_

Round 1

Reviewer 1 Report

In the presented study, Hyojin et.al. investigated the effect of Pyogallol in combating liver tumor by using hepatoma cell lines and further examined the mechanism of its action and proposed pyogallol functions on cell cycle arrest via miR134. Overall the finding is interesting and presentation is clear. However, the data is preliminary in concluding miR134 mediated its function.

Main point

The western blot data of Pyrogallol affect c-Myc, Skp2, p27 is clear. But the immunoprecipitation data of Pyrogallol disturbing interaction between those proteins are not convincing. Changes are minimal. Figure6. Compared to the WB in Fig4, changes of p27, c-Myc, Cyclin E and Cyclin D1 by Pyrogallol seems disappeared when compare the first and last column and also the rescue effect of miR134 is not visible. How is the expression level of miR134 in hepatocytes compared to other tissue? It is essential to conduct animal studies to prove if pyrogallol can selectively targeting tumor cells but not healthy cells.

Author Response

Reviewer: 1

Comments and Suggestions for Authors

In the presented study, Hyojin et.al. investigated the effect of Pyogallol in combating liver tumor by using hepatoma cell lines and further examined the mechanism of its action and proposed pyogallol functions on cell cycle arrest via miR134. Overall the finding is interesting and presentation is clear. However, the data is preliminary in concluding miR134 mediated its function.

Main point

-The western blot data of Pyrogallol affect c-Myc, Skp2, p27 is clear. But the immunoprecipitation data of Pyrogallol disturbing interaction between those proteins are not convincing. Changes are minimal. Figure6. Compared to the WB in Fig4, changes of p27, c-Myc, Cyclin E and Cyclin D1 by Pyrogallol seems disappeared when compare the first and last column and also the rescue effect of miR134 is not visible.

(Response) Thanks. I agree with you on some points. But in Fig 4, Pyrogallol disturbed interaction between c-Myc and p27 and the expression levels of blots were quantified. Also, we performed Western blot for c-Myc and new blot was added in Fig. 6. Here we can find some clear difference between 1 st and 4th column in Fig 4, while rescue effect of miR134 was visible in c-Myc and cyclin E rather than p27 or cyclin D1 in Fig. 6.

-How is the expression level of miR134 in hepatocytes compared to other tissue?

(Response) Thanks. miR134 was highly expressed in lung tumor, pancreatic cancer, colon cancer, and prostate cancer while that was low expressed in glioblastomas, breast cancer, renal cell carcinoma, colorectal cancer, hepatocellular carcinoma, osteosarcoma cell lines. (miR-134: A Human Cancer Suppressor? Mol Ther Nucleic Acids. 2017 Mar 17;6:140-149.)

-It is essential to conduct animal studies to prove if pyrogallol can selectively targeting tumor cells but not healthy cells.

(Response) Thanks for your critical comments. Due to limited time for scaling up pyrogallol for animal study, we will perform animal study with pyrogallol in the near future.

Reviewer 2 Report

ijms-564657-Comments

This manuscript described the antitumor effects of Pyrogallol on two hepatocellular carcinoma cell lines (Hep3B and Huh7) using different molecular biological techniques. In view of the writing style, this manuscript is very simple and easy understood. This is an interesting study. However, some crucial problems or errors are existed. In my opinion, this manuscript can be accepted after major revision. The authors must carefully address these suggestions listed as below,

1)    The “introduction” section is very inadequate, the authors should add some recently published articles related to novel cancer therapies (e.g., combination chemotherapy) (Future Medicinal Chemistry, 2017, 9, 403-435.; Future Medicinal Chemistry, 2018, 10, 1971-1996) and small molecule compounds, for example, 3-bromopyruvate, as potent candidate drugs of hepatocellular carcinoma (Cancers, 2019, 11, 317).

2)    line 29-30, Page 1. “Primary liver cancer is recognized the sixth most common cancer and the second leading cause of cancer mortality all over the world.” The authors must carefully check the expression, and references should be added.

3)    line 35-37, Page 1. “Among them, many natural compounds such as lupeol[6], Biochain A[7], dimethyl sulfoxide[8], sulphorafane[9] are attractive targeting S phase arrest.” Dimethyl sulfoxide is a kind of natural compound? The authors must be confused. The authors must carefully read the associated references and revise this sentence.

4)    line 58-59, Page 2. “Figure 1. Pyrogallol decreases the viability of Hep3B and Huh7 cells. (A) Chemical structure of Pyrogallol. (B) Cell were seeded in a 96-well plate and exposed……”. “Decreases” and “cell” should be changed to “decreased” and “cells”.

5)    line 75, Page 3. “…also reduced the fluorescent expression of Cyclin E in Hep3B and Huh7 cells (Fig. 3B)…”. In fact, no significant difference of the expression of Cyclin E was found after exposure to Pyrogallol in Huh7 cells. I suggest the authors improve the quality of this Figure.

6)    In the section of “Discussion”, the authors should also introduce the related results of Pyrogallol from other groups rather than repeatedly described your own results, then compared and analyzed these results from yourselves and others.

7)    I cannot understand why the antitumor effect of Pyrogallol is mainly mediated by miR134? No logical evidence can be found in your manuscript. MiR134 activation is only one factor for its antitumor activity in your current manuscript. I cannot conclude that S phase arrest and inhibition of PI3K/AKT/Skp2/cMyc signaling are mediated by miR134 activation. The authors should revise the presentation about the antitumor mechanism of Pyrogallol.

8)    Line 166-167, Page 7. “…antibodies for Skp2, CyclinE, CyclinA, p27(Santa cruz Biotechnology, Dallas, USA), (sigma, St. Louis, USA)…”. Which antibodies were purchased from sigma, St. Louis, USA?

9)    Line 185, Page 8. “(0. 20, 40, 80 μM) or (0, 20, 40, 80 μM)?”

10) Line 196, Page 8. “cDAN or cDNA?”

11) Line 222-223, Page 9. “The cell were seeded in 6well plate (1 ×105/well) and transfected with 40 nM of Inhibitor control, mimic control, miR134 inhibitor by using X-tremeGENE HP DNA Transfection reagent(Roche, Basel, Switzerland).” I don’t understand this sentence.

12) In addition, the manuscript should be polished by an English native speaker. The style of references should be carefully checked, such as [3], [8], [20].

Author Response

Reviewer: 2

Comments and Suggestions for Authors

ijms-564657-Comments

This manuscript described the antitumor effects of Pyrogallol on two hepatocellular carcinoma cell lines (Hep3B and Huh7) using different molecular biological techniques. In view of the writing style, this manuscript is very simple and easy understood. This is an interesting study. However, some crucial problems or errors are existed. In my opinion, this manuscript can be accepted after major revision. The authors must carefully address these suggestions listed as below,

 1)   The “introduction” section is very inadequate, the authors should add some recently published articles related to novel cancer therapies (e.g., combination chemotherapy) (Future Medicinal Chemistry, 2017, 9, 403-435.; Future Medicinal Chemistry, 2018, 10, 1971-1996) and small molecule compounds, for example, 3-bromopyruvate, as potent candidate drugs of hepatocellular carcinoma (Cancers, 2019, 11, 317).

(Response) Thanks. Above references were cited in Introduction.

2)    line 29-30, Page 1. “Primary liver cancer is recognized the sixth most common cancer and the second leading cause of cancer mortality all over the world.” The authors must carefully check the expression, and references should be added.

(Response) Thanks. Corrected as “Liver cancer is recognized the sixth most common cancer and the fourth leading cause of cancer mortality all over the world .”

3)    line 35-37, Page 1. “Among them, many natural compounds such as lupeol[6], Biochain A[7], dimethyl sulfoxide[8], sulphorafane[9] are attractive targeting S phase arrest.” Dimethyl sulfoxide is a kind of natural compound? The authors must be confused. The authors must carefully read the associated references and revise this sentence.

(Response) Sorry for making you confused. It was removed.

4)    line 58-59, Page 2. “Figure 1. Pyrogallol decreases the viability of Hep3B and Huh7 cells. (A) Chemical structure of Pyrogallol. (B) Cell were seeded in a 96-well plate and exposed……”. “Decreases” and “cell” should be changed to “decreased” and “cells”.

(Response) Thanks. Corrected.

5)    line 75, Page 3. “…also reduced the fluorescent expression of Cyclin E in Hep3B and Huh7 cells (Fig. 3B)”. In fact, no significant difference of the expression of Cyclin E was found after exposure to Pyrogallol in Huh7 cells. I suggest the authors improve the quality of this Figure.

(Response) Thanks. The quality of Figure was improved based on your comments..

6)    In the section of “Discussion”, the authors should also introduce the related results of Pyrogallol from other groups rather than repeatedly described your own results, then compared and analyzed these results from yourselves and others.

(Response) Thanks. We added detailed discussion compared with previous evidences.

7)    I cannot understand why the antitumor effect of Pyrogallol is mainly mediated by miR134? No logical evidence can be found in your manuscript. MiR134 activation is only one factor for its antitumor activity in your current manuscript. I cannot conclude that S phase arrest and inhibition of PI3K/AKT/Skp2/cMyc signaling are mediated by miR134 activation. The authors should revise the presentation about the antitumor mechanism of Pyrogallol.

(Response) Thanks for your critical comments. However, we suggested that miR134 activation mediates antitumor effect of Pyrogallol at least in part, which requires further study on detailed mechanisms of Pyrogallol in the future in Discussion section, since we confirmed that Pyrogallol activated expression of miR134 and conversely inhibition of miR134 rescued c-Myc and cyclin E rather than p27 or cyclin D1 in Pyrogallol treated Huh 7 cells as shown in Fig. 6.

8)    Line 166-167, Page 7. “…antibodies for Skp2, CyclinE, CyclinA, p27(Santa cruz Biotechnology, Dallas, USA), (sigma, St. Louis, USA)…”. Which antibodies were purchased from sigma, St. Louis, USA?

(Response) Sorry for inconveniences. Corrected.

9)    Line 185, Page 8. “(0. 20, 40, 80 μM) or (0, 20, 40, 80 μM)?”

(Response) Thanks. Corrected.

10) Line 196, Page 8. “cDAN or cDNA?”

(Response) Thanks. Corrected.

11) Line 222-223, Page 9. “The cell were seeded in 6well plate (1 ×105/well) and transfected with 40 nM of Inhibitor control, mimic control, miR134 inhibitor by using X-tremeGENE HP DNA Transfection reagent(Roche, Basel, Switzerland).” I don’t understand this sentence.

(Response) Thanks. Corrected.

12) In addition, the manuscript should be polished by an English native speaker. The style of references should be carefully checked, such as [3], [8], [20].

(Response) Thanks. English was carefully polished and reference style was checked as based on your comment.

Round 2

Reviewer 2 Report

The quality of this manuscript in current version has been improved compared to original manuscript. In my opinion, this manuscript can be accepted after minor revision. However, some errors or the writing should be carefully checked and revised. 

Some suggestions:

1) Line 82, Page 4, in the caption of figure 4, ".....(B) cell were....." should be revised as "....(B) Cells were...."

2) Figure 6 (a), the caption of vertical coordinate should be revised.

3) Some fundamental format errors, such as space should be added or deleted in appropriate place.

4) The style of many references is not conformed to the standard of IJMS, the authors must carefully revise them.

Author Response

Revision Note

Comments and Suggestions for Authors

The quality of this manuscript in current version has been improved compared to original manuscript. In my opinion, this manuscript can be accepted after minor revision. However, some errors or the writing should be carefully checked and revised. 

Some suggestions:

1) Line 82, Page 4, in the caption of figure 4, ".....(B) cell were....." should be revised as "....(B) Cells were...."

(Response) Thanks. Corrected.

2) Figure 6 (a), the caption of vertical coordinate should be revised.

(Response) Thanks. Corrected as “Relative miR134 mRNA level (miR134/GADPH, fold)”

3) Some fundamental format errors, such as space should be added or deleted in appropriate place.

(Response) Thanks. Carefully corrected.

4) The style of many references is not conformed to the standard of IJMS, the authors must carefully revise them.

(Response) Thanks. Reference style was checked as based on your comment.